# An atlas of polygenic risk score associations to highlight putative causal relationships across the human phenome

Tom G Richardson*, Sean Harrison, Gibran Hemani, George Davey Smith

MRC Integrative Epidemiology Unit (IEU), Population Health Sciences, Bristol Medical School, University of Bristol, Bristol, United Kingdom

**Abstract** The age of large-scale genome-wide association studies (GWAS) has provided us with an unprecedented opportunity to evaluate the genetic liability of complex disease using polygenic risk scores (PRS). In this study, we have analysed 162 PRS ($p < 5 \times 10^{-05}$) derived from GWAS and 551 heritable traits from the UK Biobank study (N = 334,398). Findings can be investigated using a web application (http://mrcieu.mrsoftware.org/PRS_atlas/), which we envisage will help uncover both known and novel mechanisms which contribute towards disease susceptibility. To demonstrate this, we have investigated the results from a phenome-wide evaluation of schizophrenia genetic liability. Amongst findings were inverse associations with measures of cognitive function which extensive follow-up analyses using Mendelian randomization (MR) provided evidence of a causal relationship. We have also investigated the effect of multiple risk factors on disease using mediation and multivariable MR frameworks. Our atlas provides a resource for future endeavours seeking to unravel the causal determinants of complex disease.
DOI: https://doi.org/10.7554/eLife.43657.001

*For correspondence:
Tom.G.Richardson@bristol.ac.uk

Competing interests: The authors declare that no competing interests exist.

## Introduction

Developing our understanding of how modifiable social, behavioural and physiological factors influence risk of disease is of vital importance to improve effective medical treatment and preventative interventions (*Abraham et al., 2016*). Genetic factors may also contribute substantially to disease susceptibility, as demonstrated by recent large-scale genome-wide association studies (GWAS) which have uncovered thousands of trait-associated single nucleotide polymorphisms (SNPs) throughout the human genome. However, typically the magnitude of effect and variance explained by one of these common genetic variants is small (*Visscher et al., 2017*). Polygenic risk scores (PRS), commonly defined as the sum of trait-associated SNPs weighted by their effect sizes, harness findings from GWAS to provide an overall measure of an individual's genetic liability to develop disease (*Torkamani et al., 2018*). Although early applications of PRS were found to be underwhelming in terms of disease prediction (*Ripatti et al., 2010*), breakthroughs in the scale of GWAS and accessibility to biobank scale datasets have substantially improved their performance (*Khera et al., 2018*; *Lee et al., 2018*). As such, they hold considerable potential to improve early disease prognosis and treatment plan formulation (*Lewis and Vassos, 2017*).

Along with the emerging utility of PRS to predict disease, they have also been previously used to evaluate putative causal relationships (*Davies et al., 2018*; *Palmer et al., 2012*). For example, instead of using a coronary heart disease (CHD) PRS to predict incidence of this disease, studies have investigated whether scores for known risk factors, such cholesterol and lipid levels (*Holmes et al., 2015*), are also strongly associated with CHD incidence. One such approach in this paradigm is Mendelian randomization (MR), a method by which genetic variants are leveraged as instrumental variables to investigate causal relationships between modifiable risk factors and disease

**eLife digest** An individual's risk of developing many diseases, including heart disease and schizophrenia, is influenced by a complex combination of lifestyle factors and the genes they inherit at birth. The total number of genetic variants that an individual has that increases their risk of developing a particular disease can be measured as their 'polygenic risk score'. These scores allow researchers to predict whether it is likely that someone will develop a disease during their lifetime.

Polygenic risk scores can also be used to link different conditions or traits to each other. For example, if high blood pressure can be caused by obesity, then genetic variants linked to obesity will also influence blood pressure. As a result, individuals with a high polygenic risk score for obesity will, on average, have a higher blood pressure than those with a low score. Comparing associations between polygenic risk scores and traits can therefore suggest whether one trait causes another.

Richardson et al. have developed an 'atlas' that uses data from the UK Biobank study – which contains genetic data from over 300,000 people – to investigate how shared characteristics and risk factors in individuals relate to their genetic likelihood of developing a disease. The data currently includes 162 different polygenic risk scores and 551 traits.

Richardson et al. used the atlas to evaluate which traits are most strongly linked to the polygenic risk score for schizophrenia. Analyses of these traits suggested that individuals with a high genetic risk of developing schizophrenia tend to perform worse in IQ and short-term memory tests, and that they are less likely to successfully quit smoking. These characteristics have previously been observed in studies of individuals with schizophrenia.

In the future, the atlas could be used to identify possible relationships between a wide range of individual traits and diseases. This could help to prioritise which relationships should be investigated further as part of studies to understand the causes and consequences of disease. In the long term, such studies should improve our ability to prevent and treat many different medical conditions.

DOI: https://doi.org/10.7554/eLife.43657.002

outcomes (*Davey Smith and Ebrahim, 2003*; *Davey Smith and Hemani, 2014*). MR is typically limited to using SNPs which survive conventional GWAS corrections (i.e. $p < 5 \times 10^{-08}$), which may lack statistical power if these variants do not explain a large proportion of trait variance. In contrast, PRS derived using a more lenient threshold (e.g. $p < 5 \times 10^{-05}$) can help recover some of this missing heritability due to a larger number of SNPs being included. This may help improve detection rates for causal relationships, which can be particularly useful when evaluating associations between genetic liability for a given trait and hundreds of diverse health outcomes. Such endeavours are commonly referred to as phenome-wide association studies (*Denny et al., 2013*; *Fritsche et al., 2018*; *Krapohl et al., 2016*; *Millard et al., 2015*).

To investigate this we undertook a preliminary simulation study to compare the performance of using a PRS to detect causal relationships with a popular MR approach (the inverse variance weighted (IVW) method (*Burgess et al., 2013*)) (*Figure 1*). Results indicated that, although using a PRS provides higher statistical power, it also suffers from substantive false positive rates due to horizontal pleiotropy, the phenomenon whereby a gene influences multiple traits via independent biological pathways (*Davey Smith and Hemani, 2014*). SNPs which are known to be pleiotropic with large effects on different and diverse traits have been found to distort findings from PRS analyses (*Felsky et al., 2018*). As a consequence, findings from phenome-wide association studies using a PRS may be useful in terms of highlighting putative causal associations, although robust evaluations are necessary to investigate results. We therefore propose using various sensitivity analyses developed in the field of MR to discern whether PRS associations represent causal relationships or not. To facilitate such future analyses, an accessible resource to evaluate associations between disease genetic liability and complex traits from across the human phenome should prove to be of considerable value.

In this study, we have constructed 162 different PRS (based on $p < 5 \times 10^{-05}$) using findings from large-scale GWAS and evaluated their association with 551 traits in up to 334,398 individuals enrolled in the UK Biobank study (*Bycroft et al., 2018*; *Sudlow et al., 2015*). To disseminate these findings, we have developed a web application to examine and visualise this derived atlas of

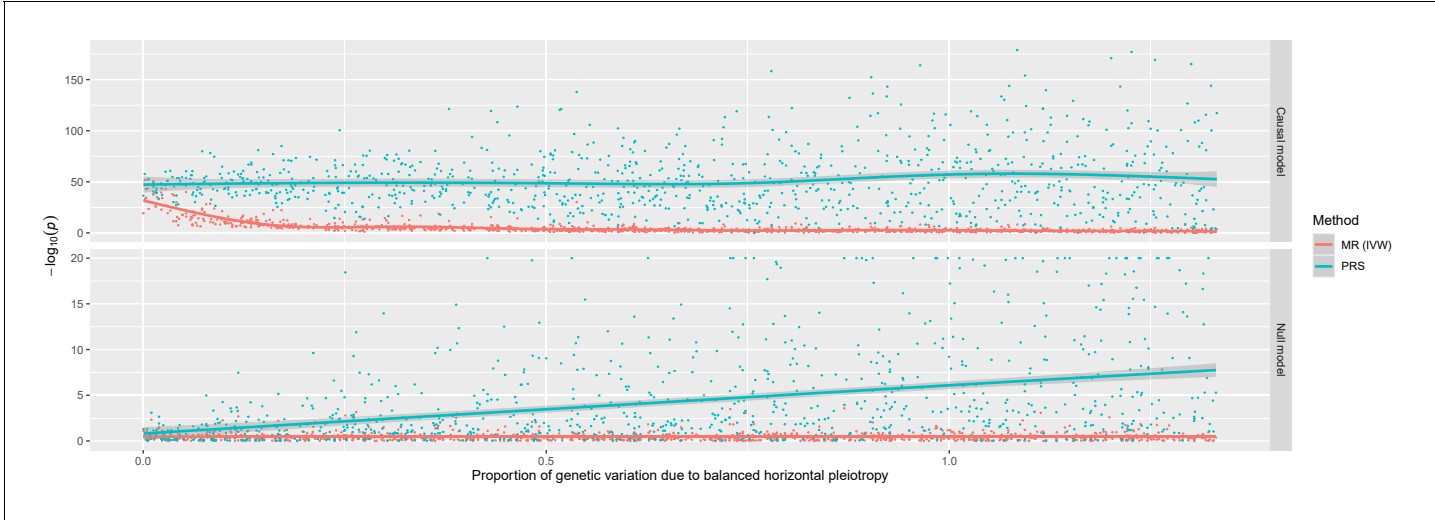

**Figure 1.** A comparison of the performance between the inverse variance weighted (IVW) Mendelian randomization (MR) model against polygenic risk score (PRS) analysis. Simulations were conducted under different levels of horizontal pleiotropy for two different models; the causal model (where the simulated exposure has a causal effect on the outcome) and the null model (where there is no causal effect between exposure and outcome).
DOI: https://doi.org/10.7554/eLife.43657.003

associations. We have also undertaken follow-up analyses to demonstrate the usefulness of this resource to help identify putative causal relationships. Firstly, we have interpreted findings from a hypothesis-free scan of associations between the schizophrenia PRS and each of the 551 traits. We demonstrate that amongst these findings are associations which may likely reflect underlying causal relationships. We have also showcased the utility of evaluating the association between all 162 PRS and a single outcome using our atlas. Using gout susceptibility as an example, we demonstrate how recently developed methodology (mediation MR and multivariable MR) can be applied to evaluate the effects of multiple risk factors on disease risk.

## Results

### An atlas of polygenic risk score associations across the human phenome
Overall, we undertook 89,262 tests to investigate the association between 162 different PRS derived from GWAS (*Supplementary file 1a*) and 551 complex traits from the UK Biobank study (*Supplementary file 1b*). PRS were constructed using independent SNPs for each GWAS ($p<5\times10^{-05}$) based on $r^2< 0.001$ using genotype data from European individuals (CEU) from phase 3 (version 5) of the 1000 Genomes project (*Abecasis et al., 2012*). As opposed to the conventional GWAS cut-off of $p<5\times10^{-08}$, the threshold of $p<5\times10^{-05}$ was selected to incorporate additional SNPs into scores which may explain additional heritability for GWAS traits. Furthermore, this allowed us to create PRS for traits which had no SNPs surviving conventional GWAS corrections, as well as increasing the number of SNPs used in scores for traits with only a small number of GWAS hits. Our final sample size for analysis consisted of 334,398 individuals. This was determined using a strict exclusion criterion to reduce false positive associations, removing individuals with withdrawn consent, evidence of genetic relatedness or who were not of 'white European ancestry' based on a K-means clustering (K = 4).

Of the 162 GWAS we identified, 11 reported that they included UK Biobank participants in their analysis. As this may lead to overfitting, the PRS for these 11 traits were not weighted to reduce this source of bias. To demonstrate this, we evaluated the association of the sleep duration PRS in the UK Biobank study, weighting SNPs based on a GWAS involving the interim release of this dataset (*Jones et al., 2016*) (*Supplementary file 1c*). However, this only mitigates this limitation, and as such these scores in particular require extensive follow-up analyses. In case they are still useful for

follow-up analyses despite overlapping with UK Biobank, these scores have been clearly flagged in *Supplementary file 1a* by being allocated to the 'unweighted' subcategory.

In this study we have only interpreted findings from associations with PRS derived using the $p<5\times10^{-05}$ threshold. However, analyses have been repeated using scores derived using the conventional GWAS threshold of $p<5\times10^{-08}$ for future studies that wish to evaluate these results. Complex traits from the UK Biobank study were selected based on p<0.05 from previously undertaken heritability analyses within this study (*Neale Lab, 2017*). This threshold was chosen as a heuristic to highlight associations worth pursuing in further detail. A web app to query and visualise these results can be found at http://mrcieu.mrsoftware.org/PRS_atlas/.

Stratifying the UK Biobank sample into deciles based on their PRS supported previous findings in the literature demonstrating the ability of PRS to predict risk of disease. For example, comparing the highest and lowest deciles of the coronary heart disease (CHD) PRS found that individuals had increased odds of 3.64 to develop this disease (based on the ICD10 code 'I25'). A recent study by Khera and colleagues (*Khera et al., 2018*) reported a similar odds ratio for CHD in their analysis (OR:>3.0 for the highest 8% of individuals based on their PRS). However, we note that they identified a higher area under curve in their analysis (0.806), which is likely attributed to tuning parameters such as LD clumping, along with covariates adjusted for in their analysis.

Combining this PRS with scores for established causal risk factors for CHD suggested that they can help improve polygenic prediction (namely low density lipoprotein (LDL) cholesterol and myocardial infarction), although integrating any associated scores in a hypothesis-free manner may hinder prediction (*Figure 2*). This could potentially due to the increase in variance incorporated into prediction analyses from scores that do not directly influence CHD, or alternatively may indicate that they are spurious associations. Additional research is required to evaluate the contribution of multiple PRS as predictors of a single outcome. Doing so may help develop a greater understanding regarding which traits can help predict disease outcomes using PRS.

Amongst other findings, we observed that participants had increased odds of 2.43 in terms of obtaining a University or College degree when comparing top and bottom deciles for the years of schooling PRS. Other noteworthy examples included a 3.48 fold increase in odds of taking atorvastatin as medication when comparing the extreme deciles for the LDL PRS. We also observed that participants in the highest decile for the ulcerative colitis PRS had increased odds of 5.36 in terms of developing this disease in comparison to those in the lowest decile (based on the ICD10 code 'K51').

## Uncovering known and novel findings by conducting a phenome-wide evaluation of associations

To demonstrate the value of this atlas of results, we have investigated some of the strongest associations detected between the schizophrenia PRS and all 551 complex traits analysed in the UK Biobank study (*Figure 3*, *Supplementary file 1d*). Associations within our atlas could potentially be identified due to underlying epidemiological relationships, although there are various other possible explanations such as a shared genetic aetiology between traits. To investigate this for our associations with the schizophrenia PRS, we have used various methods in two-sample MR as an example of how future studies could evaluate findings from our atlas. For these analyses we only used SNPs with $p<5\times10^{-08}$ as instrumental variables to reduce the likelihood of weak instrument bias in our analysis (*Davies et al., 2015*). Furthermore, in these analyses we model liability to schizophrenia as our exposure within an MR framework with associated complex traits as outcomes (unless stated otherwise). Our systematic approach involved the following:

1. As an initial evaluation, we investigated evidence of association using the inverse variance weighted (IVW) (*Burgess et al., 2013*) method and derived Cochran's Q statistic as an indicator of potential heterogeneity. Weak evidence of association in this analysis suggests that a causal effect is unlikely.
2. If the IVW method provides strong evidence of association but in the presence of heterogeneity, we suggest undertaking two additional MR analyses using the weighted mode (*Hartwig et al., 2017*) and weighed median (*Bowden et al., 2016*) methods. If there is a lack of strong evidence in both of these analyses then associations are unlikely to be causal.
3. As a sensitivity analysis, repeat steps 1 and 2 but only using SNPs as instruments which are not filtered out by applying the MR directionality test (*Hemani et al., 2017*). We also recommend

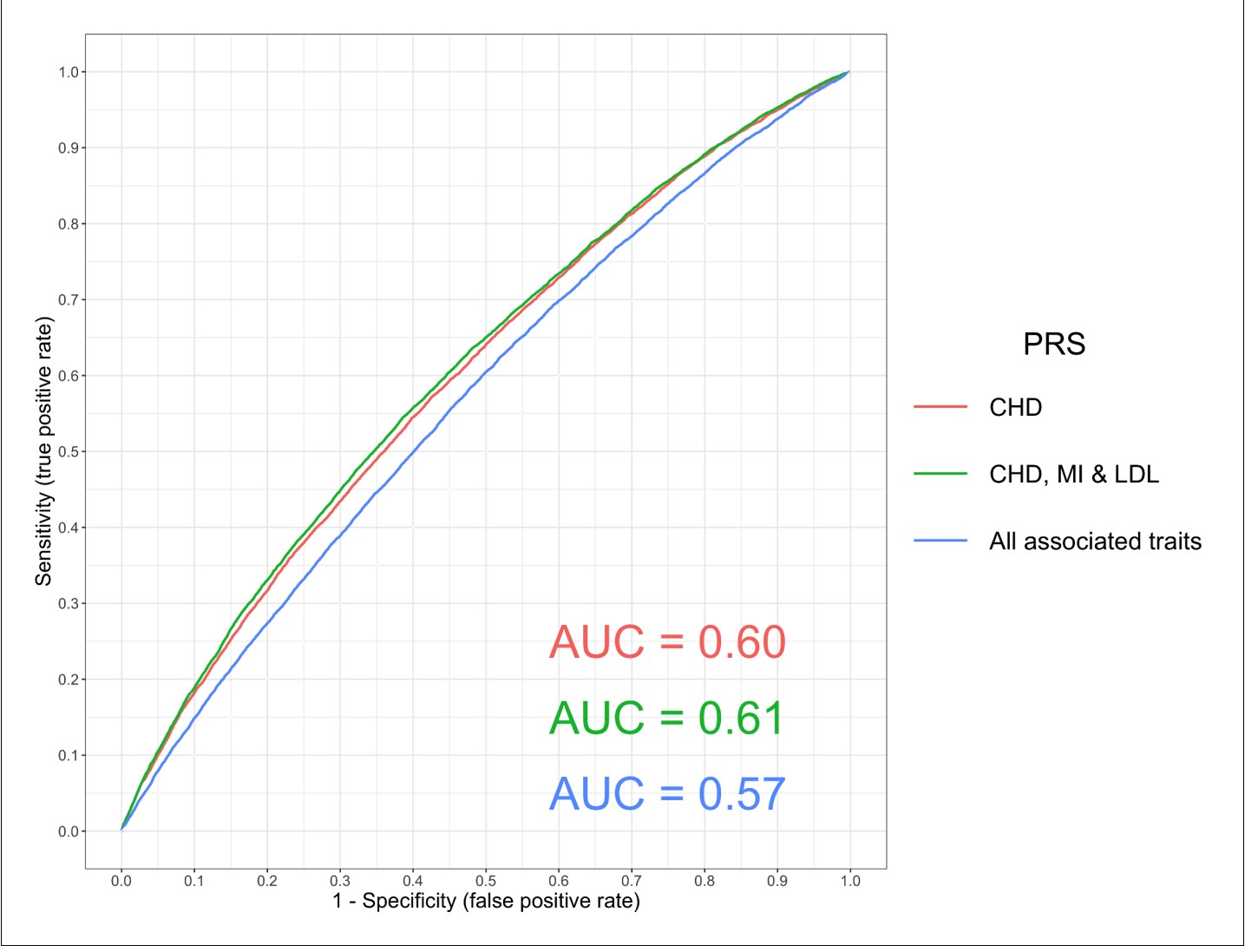

**Figure 2.** A receiver operator curve for ischaemic heart disease polygenic prediction. A receiver operating characteristic (ROC) curve to compare the sensitivity and specificity of polygenic risk scores (PRS) and individuals with ischaemic heart disease (defined using ICD 10 codes 'I25') in the UK Biobank study. The scores evaluated were 1. Coronary Heart Disease (CHD), 2. A combined scored of CHD, Myocardial Infarction (MI) and Low Density Lipoprotein cholesterol (LDL), 3. All traits with a p-value<1×10$^{-06}$ in our PRS analysis (excluding scores from GWAS overlapping with the UK Biobank sample). These were CHD, MI, LDL, Total cholesterol, Triglycerides, High Density Lipoprotein cholesterol, Years of schooling, Height and Waist Circumference. All PRS were constructed from GWAS using independent SNPs with p<5×10$^{-05}$.

DOI: https://doi.org/10.7554/eLife.43657.004

evaluating the MR-Egger intercept term (*Bowden et al., 2015*) to discern whether estimates may be biased by directional pleiotropic effects.

The top association with the schizophrenia PRS suggests that individuals with high schizophrenia genetic liability have increased odds of seeing a psychiatrist at some point in their lives due to nerves, anxiety, tension of depression (OR = 1.09 per standard deviation increase in PRS, 95% CI = 1.08 to 1.10, p=1.55×10$^{-50}$). The schizophrenia PRS was also strongly associated with various neurological traits, such as neuroticism (Beta = 0.066, SE = 0.006, p=8.17×10$^{-27}$), being 'tense or highly strung' (OR = 1.07, 95% CI = 1.07 to 1.08, p=2.25×10$^{-47}$) and self-reported depression (OR = 1.07, 95% CI = 1.06 to 1.08, p=4.91×10$^{-18}$).

We identified strong evidence that schizophrenia genetic liability influences this set of neurological traits (*Supplementary file 1e*), except for self-reported depression where strong evidence was only detected using the inverse variance weighted (IVW) method (Beta = 0.004, SE = 0.001,

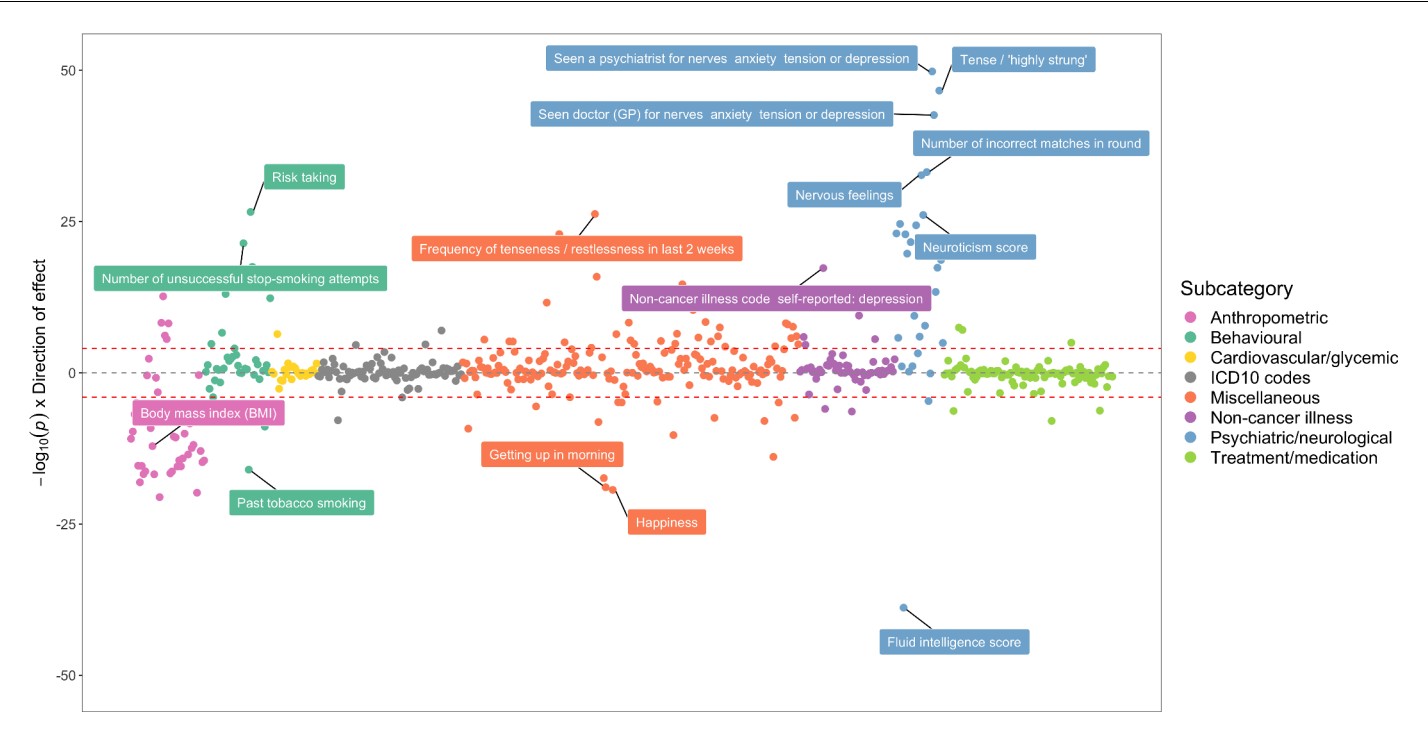

**Figure 3.** A bi-directional phenome-wide association plot for schizophrenia genetic liability. Each point on this plot represents the association between the schizophrenia polygenic risk score (based on p<5×10$^{-05}$) and a complex trait in the UK Biobank study. Along the y-axis are –log10 p-values for these associations multiplied by the direction of effect for their corresponding effect size. As such, traits positively associated with schizophrenia genetic liability reside above the horizontal grey line representing the null (i.e. –log10 (P) = 0), whereas negative associations are below. Points are grouped and coloured based on their corresponding complex traits' subcategory. Horizontal red lines indicate the Bonferroni corrected threshold for the 551 tests undertaken (i.e. 0.05/551 = 9.07×10$^{-05}$).

DOI: https://doi.org/10.7554/eLife.43657.005

p=0.009). There was also no strong evidence of directional horizontal pleiotropy for these results based on the MR Egger intercept term and associations were detected after repeating analyses using MR directionality filtering.

Along with using MR to investigate the effect of PRS traits on outcomes, we recommend investigating the converse direction of effect where possible (also known as 'bi-directional' MR (**Timpson et al., 2011**). For example, for the associations detected with the schizophrenia PRS, associated traits in the UK Biobank were modelled as our exposure in an MR setting and schizophrenia was treated as our outcome. Results suggested that neuroticism liability influences schizophrenia risk (**Supplementary file 1f**), although we detected evidence of directional horizontal pleiotropy based on the MR Egger intercept term (Beta = 0.043, SE = 0.018, p=0.018). After applying MR directionality filtering, we also identified evidence of association between being 'tense or highly strung' and schizophrenia risk. Therefore, the most parsimonious explanation for these findings could be that they have been observed due to a shared genetic aetiology between schizophrenia and other neurological traits. This is also likely to be a plausible explanation for other associations within our atlas. In particular, caution is advised when interpreting findings between autoimmune traits which are known to be influenced by highly correlated genes residing in the HLA region of the genome (**Gough and Simmonds, 2007**). Although these findings could still be of interest in terms of genetic correlations between traits, they may not reflect underlying causal relationships (**O'Connor and Price, 2018**).

Amongst other findings, there were associations which suggested individuals with high schizophrenia genetic liability had a lower fluid intelligence score (Beta = −0.083, SE = 0.006, p=1.49×10$^{-39}$). We also observed evidence that these individuals performed worse than others in an assessment of cognitive function concerning memorising pairs of cards (Beta = 0.020, SE = 0.002, p=6.66×10$^{-34}$ for 'number of incorrect matches'). Follow-up MR analyses provided evidence from

multiple methods that schizophrenia genetic liability (i.e. our exposure) influences both of these outcomes (*Supplementary file 1g*). These results were robust to sensitivity analyses using MR directionality filtering and MR Egger intercepts did not indicate that findings were prone to directional horizontal pleiotropy. In contrast, we did not detect strong evidence of a causal effect in the opposite direction for these associations (i.e. evaluating the effect on measures of cognition and memory on schizophrenia risk), in particular after applying MR directionality filtering and when evaluating results from the weighted median and mode methods (*Supplementary file 1h*). We also conducted a leave-one out analysis which suggested that no individual SNPs were responsible for driving observed effects (*Appendix 1—Figure 1*, *Appendix 1—Figure 2*). Taken together, these analyses support evidence that schizophrenia genetic liability may lead to reduced cognitive function.

Elsewhere, there were associations indicating that participants with a high schizophrenia PRS were more likely to be unsuccessful when attempting to quit smoking (Beta = 0.028, SE = 0.003, p=3.87×10$^{-22}$) and, accordingly reduced odds of being a past smoker (OR = 0.97, 95% CI = 0.97 to 0.98, p=9.71×10$^{-17}$). We observed strong evidence of association that schizophrenia genetic liability influences these outcomes (*Supplementary file 1i*), whereas the converse direction of effect provided weak evidence of an effect (*Supplementary file 1j*). However, the 'number of unsuccessful smoking attempts' outcome could only be instrumented using a single variant which limits our ability to investigate this effect. Moreover, a recent study has uncovered a large number of SNPs robustly associated with smoking cessation and provided evidence of a bi-directional relationship between smoking and schizophrenia using MR (*Wootton, 2018*). Leave-one out analyses suggested that no individual SNP was responsible for driving observed associations (*Appendix 1—Figure 3*, *Appendix 1—Figure 4*).

We also observed a strong inverse association between the schizophrenia PRS and various anthropometric traits. However, evaluating the relationship between schizophrenia liability and body mass index (BMI) provided weak evidence of a causal effect in both directions (*Supplementary files 1k & 1l*). This result reinforces our recommendation that all findings within our atlas require in-depth evaluation to discern whether they represent potential causal associations.

## Elucidating risk factors which may play a mediating role along the causal pathway to disease

Another strength of our atlas is that findings can be evaluated by selecting an outcome of interest and evaluating which of the 162 PRS are most strong associated with it. Doing so may motivate future endeavours to investigate the effect of multiple risk factors on disease risk. As a demonstration of this, we have evaluated the associations between all PRS and self-reported gout in the UK Biobank study (*Supplementary file 1m*). In this analysis, there was strong evidence of association using the PRS for gout itself (OR = 1.16, 95% CI = 1.13 to 1.19), although we also observed a much larger magnitude of effect using the urate PRS (OR = 1.75, 95% CI = 1.72 to 1.78). Although many of the PRS in our analysis may be the best polygenic predictors for their target disease/trait, there may be other examples similar to this where the strongest association for an outcome is not the corresponding PRS. For example, the strongest association for birth weight as an outcome in our atlas was with the height PRS (Beta = 0.080, SE = 0.002, p=1.31×10-249).

A receiver operating characteristic plot (*Figure 4*) illustrates this point, where the area under curve for the gout PRS was 0.54 in comparison to the urate PRS which had a value of 0.65. This may be attributed to gout being a binary outcome heavily influenced by the number of cases analysed in its corresponding GWAS (N = 2,115). In comparison, urate is a continuous trait measured in all participants for its respective GWAS (N = 110,347). After urate, the next strongest positive associations with self-reported gout were triglycerides (TG) and body mass index (BMI) (OR = 1.14, 95% CI = 1.11 to 1.16 and OR = 1.09, 95% CI = 1.06 to 1.12 respectively). However, it is unclear whether these risk factors influence gout risk independently of one and other or if they reside on the same causal pathway to disease.

We investigated this by firstly using an MR mediation framework which involved evaluating bi-directional relationships for each risk factor in turn. As before, only SNPs with p<5×10$^{-08}$ for each PRS were used as instrumental variables in MR analyses. There was strong evidence that BMI (i.e. our exposure) had a causal effect on each other trait in turn (TG, urate and gout), where effect estimates appeared to be consistent between different MR methods (*Supplementary file 1n*). Repeating this analysis for TG as our exposure provided evidence of a causal effect on urate and gout risk,

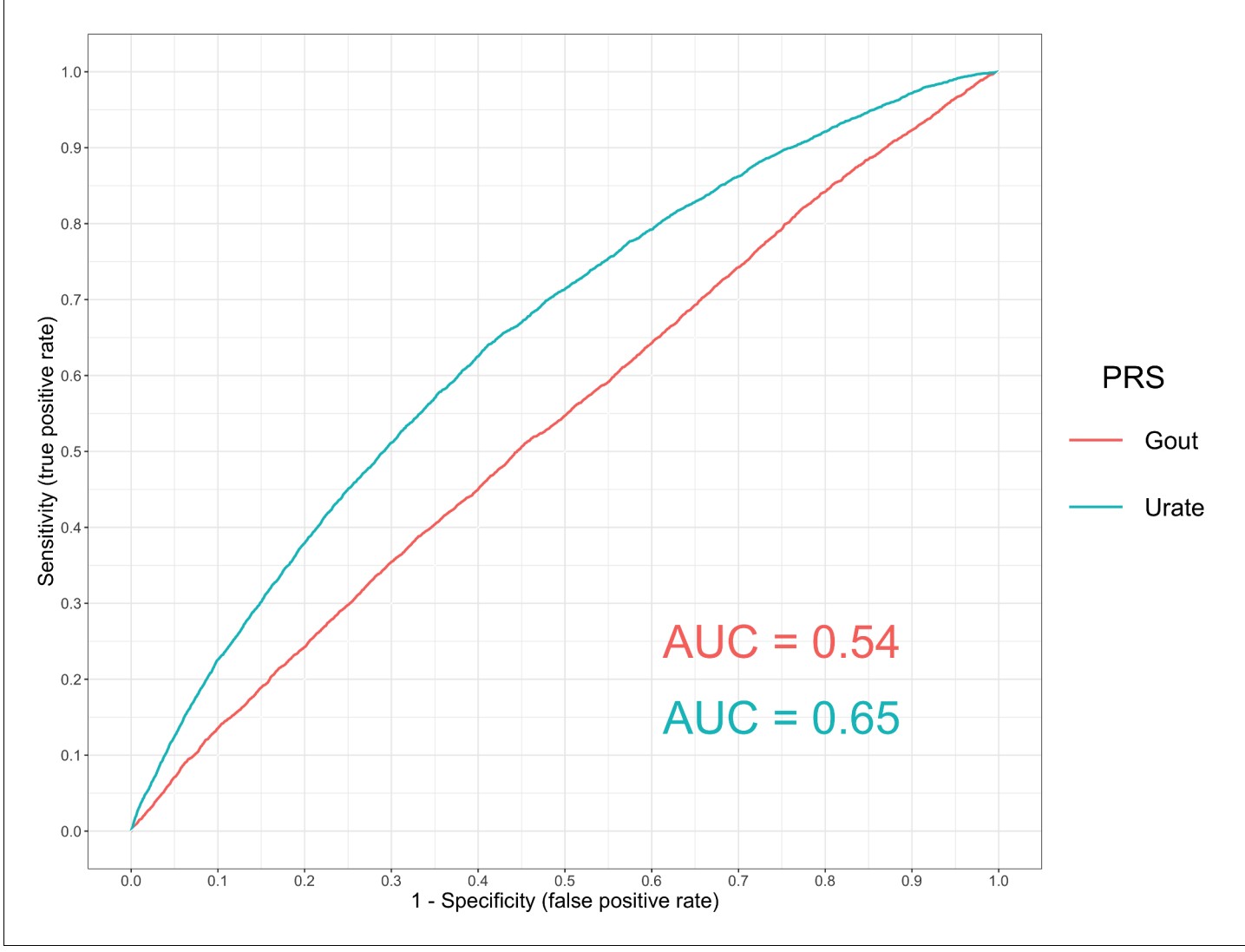

**Figure 4.** A receiver operator curve for gout polygenic prediction. A receiver operating characteristic (ROC) curve to compare the sensitivity and specificity of polygenic risk scores (PRS) and individuals with self-reported gout in the UK Biobank study. The scores evaluated were gout and urate using independent SNPs identified by GWAS with $p<5\times10^{-05}$.

DOI: https://doi.org/10.7554/eLife.43657.006

but not BMI (*Supplementary file 1o*). We then modelled urate as our exposure variable, which suggested that increased urate positively influences gout risk, although there was weak evidence of an effect on either BMI or TG (*Supplementary file 1p*). In all analyses there was no strong evidence of horizontal pleiotropy based on the MR-Egger intercept terms and findings were robust to sensitivity analyses using MR directionality filtering (*Supplementary file 1n-1p*). We also undertook leave-one out analyses which found that no single SNP was driving observed effects (*Appendix 1—Figure 5*, *Appendix 1—Figure 6*, *Appendix 1—Figure 7*, *Appendix 1—Figure 8*). In conclusion, as illustrated in *Figure 5a*, findings from the mediation MR analysis suggests that BMI influences TG levels (*Figure 5a* (1)), which has an effect of urate (*Figure 5a* (2)), and this subsequently influences gout risk (*Figure 5a* (3)). Using the effect estimates from our IVW analysis, we estimated that 77% of the overall effect of BMI on gout risk (*Figure 5a* (4)) is mediated through this causal pathway.

We also used a related approach to investigate the effect of these multiple risk factors on gout susceptibility, known as multivariable MR (*Sanderson et al., 2018*). In this analysis genetic instruments for all exposures (i.e. BMI, TG and urate) are modelled simultaneously to investigate whether these risk factors influence our outcome (i.e. gout) independently of one and other. We observed

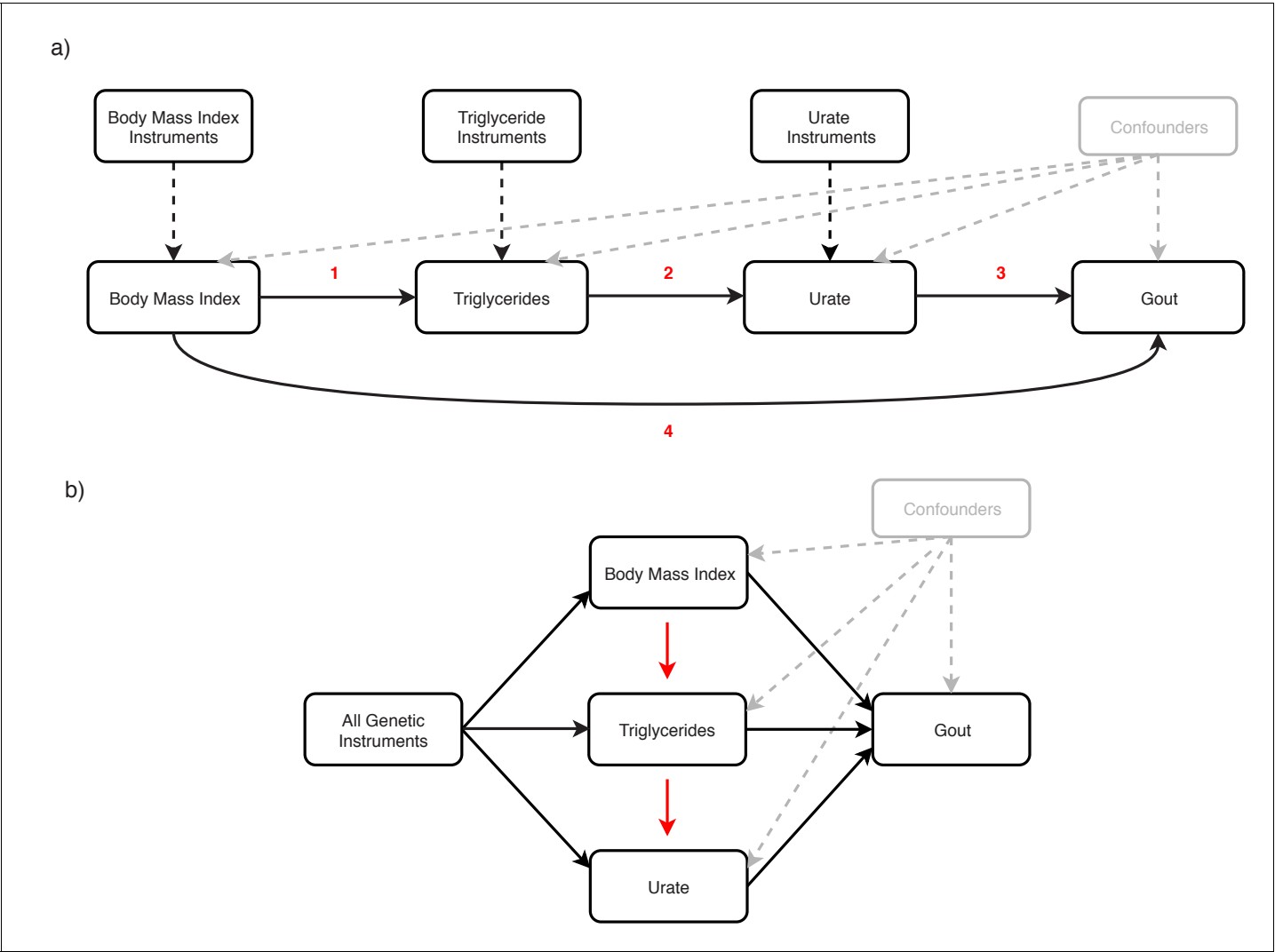

**Figure 5.** Applying (a) mediation and (b) multivariable Mendelian randomization investigate the causal effect of body mass index, triglycerides and urate on gout risk. (a) Mediation Mendelian randomization (MR) framework to investigate whether urate mediates the effect of body mass index (BMI) and triglycerides (TG) on gout risk. The various analyses undertaken suggest that 1) elevated BMI increases TG levels 2) which subsequently has an effect on urate 3) and this in turn influences gout risk. This mediation pathway may help explain the manner by which BMI, potentially driven by lifestyle factors such as diet, is a risk factor for gout. (b) Multivariable MR framework attempting to reproduce findings from the mediation analysis. Genetic instruments for BMI, TG and urate were analysed simultaneously to evaluate the joint effect of these risk factors on gout risk. The effect of BMI and TG on gout risk attenuated compared to univariable analyses, suggesting that they influence gout risk through increased urate levels. Investigating each combination of pairwise risk factors using this framework suggested that BMI influences TG rather than the opposite direction of effect, which also supports findings from the mediation analysis.

DOI: https://doi.org/10.7554/eLife.43657.007

the effects of BMI and TG on gout risk attenuate when analysed in the same model as urate (*Supplementary file 1q*). Furthermore, in subsequent analyses we applied multivariable MR to investigate each pairwise combination of these risk factors on gout risk. There was evidence of an attenuation of the effect of BMI on gout risk when accounting for either the TG or urate effect (*Supplementary files 1r and 1s*). We also observed the effect of TG on gout risk attenuate when accounting for urate levels (*Supplementary file 1t*). These findings therefore support the same direction of effect observed using the mediation framework (*Figure 5b*).

## Discussion

In this study we have developed an atlas of associations between PRS and complex traits across the human phenome. Along with contributing to mounting evidence that PRS can be valuable in predicting later life disease outcomes, we have provided examples of how this resource can be harnessed to help identify potential risk factors in disease which warrant further investigation. We envisage that the inferences we have made in this study are just the beginning of potential findings which can be uncovered using such catalogues of associations. Multiple lines of evidence from robust follow-up studies of putative causal risk factors will help improve our understanding of disease susceptibility (*Munafò and Davey Smith, 2018*).

Large-scale biobank datasets provide an unparalleled opportunity to undertake hypothesis-free causal inference. Such efforts can help identify evidence supporting established causal relationships, as well as potentially implicating novel ones (*Davey Smith and Hemani, 2014*; *Cai et al., 2018*). We have illustrated this approach in our study by evaluating the results of a phenome-wide association study of schizophrenia genetic liability. This identified strong associations with measures of cognitive function and smoking behaviour which MR follow-up analyses suggested may be due to putative causal relationships with schizophrenia genetic liability.

There is long standing evidence from the literature that cognitive impairment is a recognised characteristic of schizophrenia (*Mohamed et al., 1999*). Although PRS may prove useful in determining lifelong risk of developing schizophrenia, based on currently available data they may be less effective in terms of predicting age of schizophrenia onset as well as the severity of its progression. Characterization of cognitive decline in individuals with a high schizophrenia PRS may therefore help improve elucidation of its neurological basis, and ultimately improvement in therapeutic approaches to it (*Green, 1996*).

There is also a wealth of evidence in the literature from observational studies that individuals diagnosed with schizophrenia smoke more frequently compared to the general population (*Sacco et al., 2005*). Our results indicate that UK Biobank participants with a high schizophrenia genetic liability are more likely to be unsuccessful in their attempts to stop smoking. This may therefore suggest that the high frequency of schizophrenia patients who smoke could be attributed to their inability to quit smoking. However, we were unable to support recent evidence which suggests that smoking is a risk factor for schizophrenia which could be attributed to weak instruments in our analysis (*Wootton, 2018*). The positive association with smoking behaviour may also provide a possible explanation for the inverse association we observed between schizophrenia genetic liability and anthropometric traits.

In this study we have also provided an example of how investigating various PRS associations with the same outcome may help motivate studies evaluating the effect of multiple risk factors on disease risk. Our analysis detected evidence of an association between body mass index and gout risk, putatively mediated by triglycerides and urate levels. The findings from this analysis therefore appear to recapitulate known biology regarding the established causal pathway to gout (*Matsubara et al., 1989*), (*Li et al., 2017*). Speculatively, a diet including high calorie and alcohol consumption, which are known risk factors for increased body mass index and triglyceride levels, may result in elevated circulating uric acid level and in turn increase gout risk. A recent study has suggested that genetic factors may have a greater impact on serum urate levels than environmental factors such as diet (*Major et al., 2018*). Our findings suggest that genetic drivers of appetite which may influence higher BMI levels are likely to predominantly influence gout risk via increased urate levels. We hope this illustration will motivate creative hypotheses for future endeavours to investigate the effect of multiple risk factors on disease risk.

The application of PRS is a topic which has sparked considerable recent debate, particularly concerning whether scores are relevant for clinical decision making (*Warren, 2018*). Although resources such as the UK Biobank provide an unparalleled opportunity to investigate the determinants of complex disease as we have done in this study, findings regarding genetic liability may not be generalizable to individuals who are not of European descent. As such, there is likely to be an emphasis in the forthcoming years on efforts to establish disease-specific datasets for a diverse range of ancestries. We also note that, although we have adjusted all analyses in our study using the top 10 principal components from the UK Biobank, there may still be an influence of geographic clustering which remains unaccounted for *Abdellaoui et al. (2018)*. Furthermore, although we have flagged the PRS

traits in our study derived using GWAS which have overlapping samples with the UK Biobank, we are unable to assess this for scores whose GWAS predate this cohort. Future efforts to link anonymous identifiers between the UK Biobank and UK cohorts would be of helpful in terms of ascertaining this information to prevent overfitting. Lastly, certain complex traits in our study may benefit from being combined to improve statistical power. For instance, a more powerful approach to identify associations between genetic liability and statin medication could involve deriving a combined measure of all the different types of statins reported. Investigating these results in a hypothesis-free manner as we have described in this study may also prove useful for drug repurposing efforts.

Polygenic risk scores hold huge promise in the era of large-scale genetic epidemiology to identify individuals who are at high risk of disease. Associations detected between these scores and outcomes undertaken by large-scale analyses should prove powerful for future studies that wish to unravel causal relationships between complex traits. Doing so will help improve disease prevention by developing a stronger understanding of complex epidemiological pathways.

## Materials and methods

### Simulations to compare polygenic risk score analysis with Mendelian randomization

Our simulation study concerned two different models; the causal model (simulating a risk factor which has a causal effect on the simulated outcome) and the null model (where there is no causal effect between the simulated exposure and outcome). We ran 1000 simulations using each model to compare the PRS approach with the IVW method using a dataset comprising of 10,000 samples and 50 SNPs. Further details and all the code used to conduct these simulations can be found at https://github.com/explodecomputer/prs-vs-mr.

### Constructing polygenic risk scores from large-scale genome-wide association studies

We have used the MR-Base platform (*Hemani et al., 2018*) to identify SNPs from large-scale GWAS to include in our PRS. Our inclusion criteria for selected GWAS was having a sample size of more than 1000 participants, over 100,000 SNPs measured on genotyping arrays and based on European/ mixed populations. If multiple studies were found for the same trait, we selected the most recent study or the one with the largest sample size.

PRS were constructed using SNPs for each GWAS trait based on $p < 5 \times 10^{-05}$. A threshold of $r^2 < 0.001$ was selected to identify independent SNPs using genotype data from European individuals (CEU) from phase 3 (version 5) of the 1000 genomes project (*Abecasis et al., 2012*). When a GWAS SNP was not available from the UK Biobank study genotype data, we used a proxy SNP instead based on $r^2 \geq 0.8$ using the same reference panel. Scores were then calculated as the sum of the effect alleles for all SNPs weighted by their reported regression coefficients. However, a small subset of PRS were left unweighted to reduce the likelihood of overfitting. This was due to their GWAS including participants from the initial release of the UK Biobank study. As such, additional caution should be exercised when interpreting findings from these unweighted PRS. Prior to analysis, each PRS was normalised to have a mean of zero and a standard deviation (SD) of one. Our PRS construction pipeline was also applied using a more stringent threshold of $p < 5 \times 10^{-08}$. Although we have not interpreted any of the results using these more stringent scores in this report, they are available within our atlas for future use.

### Complex trait and genotype data from the UK Biobank study

We selected traits from the UK Biobank study (*Sudlow et al., 2015*) which had $p < 0.05$ in the heritability analyses conduct by the Neale lab (*Neale Lab, 2017*). Genotype data were available for approximately 490,000 individuals enrolled in the study. Phasing and imputation of these data are explained elsewhere (*Bycroft et al., 2018*). Individuals with withdrawn consent, evidence of genetic relatedness or who were not of 'white European ancestry' based on a $K$-means clustering ($K = 4$) were excluded from analysis. After exclusions there were up to 334,398 individuals with both genotype and complex trait data who were eligible for analysis.

## Statistical analysis

We evaluated the association between each combination of PRS and complex trait in the UK Biobank study using linear regression (for continuous traits), logistic regression (for case/control traits), ordinal logistic regression (for ordered categorical traits) and multinomial logistic regression (for unordered categorical traits). All analyses were adjusted for age, sex, the first 10 genetic principle components (to adjust for population stratification) and genotyping chip used to measure genetic data in participants. Only female participants were included in the 'Age at menarche' and 'Age at menopause' PRS analyses.

We also calculated $R^2$ coefficients for continuous traits and McFadden pseudo $R^2$ coefficients for other models by repeating analyses unadjusted for covariates. McFadden's $R^2$ is defined as:

$$R^2_{McF} = 1 - \ln(L_m)/\ln(L_0)$$

where ln is the natural logarithm, $L_0$ is the value of the likelihood function of the model with no predictors and $L_m$ is the likelihood of the model being estimated. We note that pseudo $R^2$ coefficients should not be interpreted in a similar manner to those derived using linear regression (*Hu et al., 2006*).

## Mendelian randomization analysis

We used various two-sample MR methods to evaluate associations detected in the PRS analysis. This involved using the observed effects of the genetic variants used in the PRS on both the GWAS trait that the score was based on (treated as the exposure in our MR analysis) as well as the UK Biobank trait (treated as the outcome in our MR analysis). For all MR analyses we only selected SNPs with $p < 5 \times 10^{-08}$ based on GWAS findings as instrumental variables to reduce the likelihood of weak instrument bias (*Davies et al., 2015*). In terms of MR methods, we applied the inverse variance weighted (IVW) (*Burgess et al., 2013*), weighed median (*Bowden et al., 2016*) and weighted mode approaches. We also conducted several different sensitivity analyses to evaluate findings. We derived Cochran's Q statistic when undertaking the IVW approach as an indicator of heterogeneity, as well as repeating all analyses after filtering out SNPs which the MR directionality test suggested did not influence the outcome of interest through the analysed exposure. The intercept of the MR-Egger approach (*Bowden et al., 2015*) was used to investigate directional horizontal pleiotropy and leave-one-out analyses (i.e. reapplying the IVW method after removing each SNP in turn with replacement) were conducted to discern whether any individual SNPs were driving observed associations. These types of analyses are particularly important when assessing findings from our atlas, as one possible explanation is that they could be attributed to a single pleiotropic SNP which has a large effect size (e.g. the *APOE* locus which is associated with Alzheimer's disease and lipid levels).

To investigate the direction of effect for associations identified in the PRS analysis we undertook bi-directional MR (*Timpson et al., 2011*). This involves firstly modelling our PRS trait as our exposure and complex trait as our outcome, and subsequently the complex trait as our exposure and PRS trait as our outcome in a separate analysis. Lastly, we have incorporated two recent developments within the field of MR; mediation MR and multivariable MR (*Sanderson et al., 2018*). These methods can be used to investigate the effect of multiple risk factors on a single outcome, as well as uncover potential mediators in disease. In this study we have evaluated findings from the PRS analysis based on the $p < 5 \times 10^{-05}$ threshold. We note however that it is only advisable to apply techniques in MR using this threshold as long as in-depth sensitivity analyses (e.g. leave-one out, MR-Egger intercept) are also undertaken to robustly evaluate findings.

When undertaking our example of mediation MR in this study, we also calculated the proportion mediated along the causal pathway from exposure to outcome using effect estimates derived using the IVW method, where:

$$\text{Proportion mediated} = \frac{\text{direct effect} - \text{indirect effect}}{\text{direct effect}}$$

The direct effect here is the IVW effect estimate derived for the association between the exposure (i.e. BMI) and our outcome (i.e. gout). The indirect effect was calculated as the product of all IVW effect estimates derived for all relationships along the causal pathway of interest (i.e. the effect of BMI on triglycerides, the effect of triglycerides on urate and the effect of urate on gout).

All analyses were undertaken using R (version 3.5.1). The R package 'shiny' v1.1 was used to develop the web application and 'highcharter' v0.5 was used to generate interactive plots. Figures in this manuscript were generated using 'ggplot2' v2.2.1.

## Data availability

All summary statistics for the analyses undertaken in this study can be downloaded using our web application (http://mrcieu.mrsoftware.org/PRS_atlas/). Our dataset was derived from the UK Biobank study as part of projects 8786 and 15825. The same dataset can be created with an application to use data from the UK Biobank study (http://biobank.ctsu.ox.ac.uk/crystal/).

## Acknowledgements

We are extremely grateful to all the authors of the genome-wide association studies who have made their summary statistics publicly available for the benefit of this study. We would also like to thank the efforts of the Neale Lab who conducted extensive heritability analyses in the UK Biobank which guided our selection of traits to analyse.

This work was supported by the Integrative Epidemiology Unit which receives funding from the UK Medical Research Council and the University of Bristol (MC_UU_00011/1). SH is part of a project entitled 'social and economic consequences of health: causal inference methods and longitudinal, intergenerational data', which is part of the Health Foundation's Efficiency Research Programme. The Health Foundation is an independent charity committed to bringing about better health and health care for people in the UK. GH is supported by the Wellcome Trust [208806/Z/17/Z]. TGR is a UKRI Innovation Research Fellow (MR/S003886/1).

## Additional information

### Funding

| Funder | Grant reference number | Author |
|---|---|---|
| Medical Research Council | MC_UU_00011/1 | George Davey Smith |
| Wellcome Trust | 208806/Z/17/Z | Gibran Hemani |
| Health Data Research UK | MR/S003886/1 | Tom G Richardson |

The funders had no role in study design, data collection and interpretation, or the decision to submit the work for publication.

### Author contributions

Tom G Richardson, Conceptualization, Resources, Data curation, Software, Formal analysis, Validation, Investigation, Visualization, Methodology, Writing—original draft, Writing—review and editing; Sean Harrison, Data curation, Methodology, Writing—review and editing; Gibran Hemani, Supervision, Methodology, Writing—review and editing; George Davey Smith, Conceptualization, Supervision, Methodology, Writing—review and editing

### Author ORCIDs

Tom G Richardson https://orcid.org/0000-0002-7918-2040
Gibran Hemani http://orcid.org/0000-0003-0920-1055
George Davey Smith http://orcid.org/0000-0002-1407-8314

### Ethics

Human subjects: All data used in this study has been obtained from the UK Biobank study who have obtained ethics approval from the Research Ethics Committee (REC - approval number: 11/NW/0382). All participants in enrolled this study have signed consent forms.

### Decision letter and Author response

Decision letter https://doi.org/10.7554/eLife.43657.020
Author response https://doi.org/10.7554/eLife.43657.021

## Additional files

### Supplementary files

• Supplementary file 1. Supplementary Tables. Supplementary file 1a: A list of the 162 traits and corresponding GWAS identified to construct polygenic risk scores. Supplementary file 1b: A list of the 551 complex traits analysed in the UK Biobank study. Supplementary file 1c: An evaluation of weighted and unweighted PRS where samples overlap between discovery and application. Supplementary file 1d: A phenome-wide evaluation of schizophrenia genetic liability. Supplementary file 1e: Mendelian randomization analysis to investigate the effect of schizophrenia genetic liability on neurological traits. Supplementary file 1f: Mendelian randomization analysis to investigate the effect of neurological traits on schizophrenia risk. Supplementary file 1g: Mendelian randomization analysis to investigate the effect of schizophrenia genetic liability on cognitive traits. Supplementary file 1h: Mendelian randomization analysis to investigate the effect of cognitive traits on schizophrenia risk. Supplementary file 1i: Mendelian randomization analysis to investigate the effect of schizophrenia genetic liability on smoking traits. Supplementary file 1j: Mendelian randomization analysis to investigate the effect of smoking traits on schizophrenia risk. Supplementary file 1k: Mendelian randomization analysis to investigate the effect of schizophrenia genetic liability on body mass index. Supplementary file 1l: Mendelian randomization analysis to investigate the effect of body mass index on schizophrenia risk. Supplementary file 1m: A genetic-liability phenome scan for self-reported gout. Supplementary file 1n: Mendelian randomization analysis to investigate the effect of body mass index on triglycerides, urate and gout. Supplementary file 1o: Mendelian randomization analysis to investigate the effect of triglycerides on body mass index, urate and gout. Supplementary file 1p: Mendelian randomization analysis to investigate the effect of urate on body mass index, triglycerides and gout. Supplementary file 1q: Multivariable Mendelian randomization analysis to investigate the effect of body mass index, triglycerides and urate on gout risk. Supplementary file 1r: Multivariable Mendelian randomization analysis to investigate the effect of body mass index and urate on gout risk. Supplementary file 1s: Multivariable Mendelian randomization analysis to investigate the effect of triglycerides and urate on gout risk. Supplementary file 1t: Multivariable Mendelian randomization analysis to investigate the effect of body mass index and triglycerides on gout risk.
DOI: https://doi.org/10.7554/eLife.43657.008

• Transparent reporting form
DOI: https://doi.org/10.7554/eLife.43657.009

### Data availability

All output generated in this project has been made available at: http://mrcieu.mrsoftware.org/PRS_atlas/ (a copy of the data is archived at http://dx.doi.org/10.5061/dryad.h18c66b).

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

## Appendix 1

DOI: https://doi.org/10.7554/eLife.43657.010

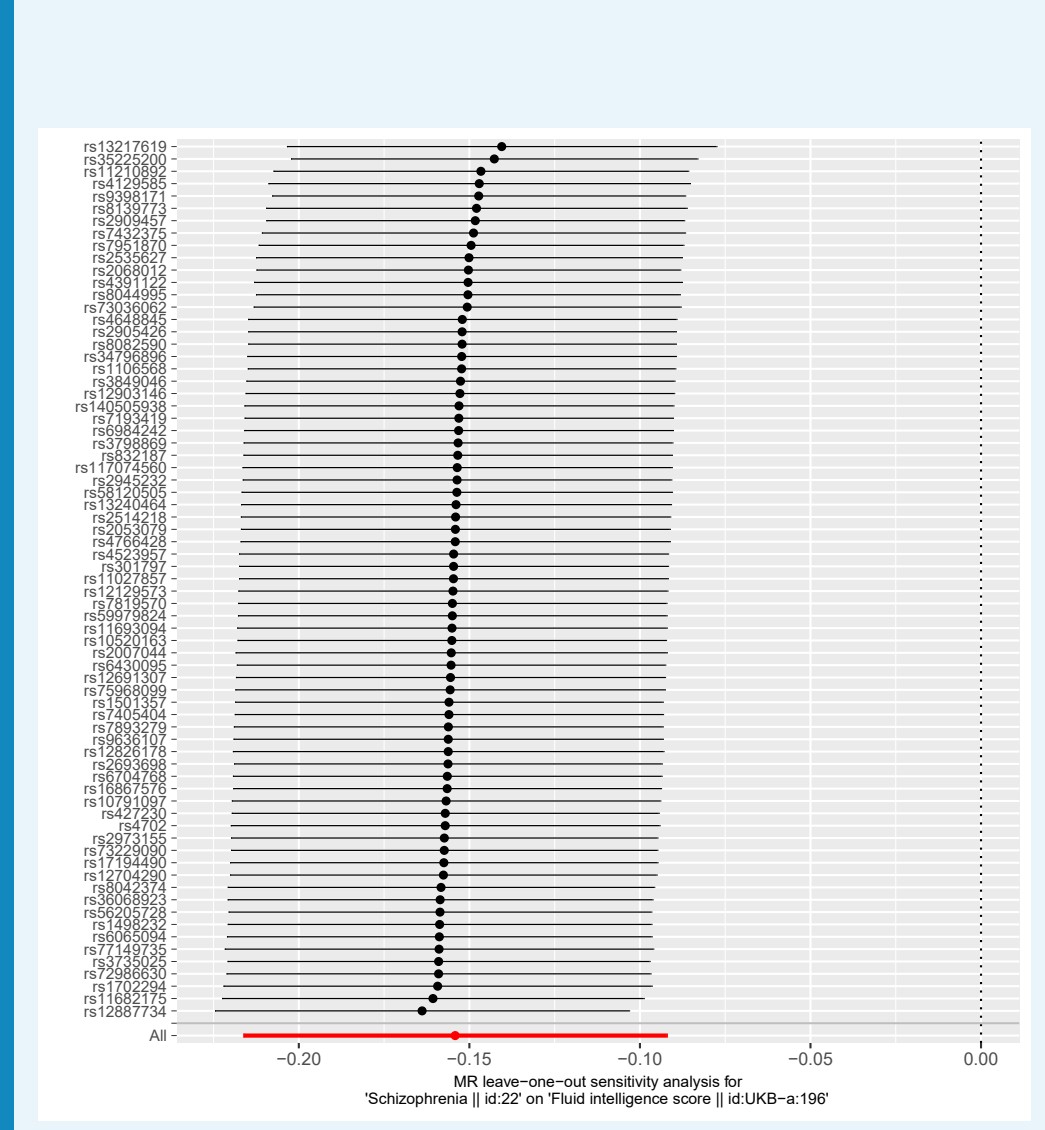

MR leave−one−out sensitivity analysis for
'Schizophrenia || id:22' on 'Fluid intelligence score || id:UKB−a:196'

**Appendix 1—figure 1.** A plot illustrating a leave-one out analysis between schizophrenia genetic liability and fluid intelligence.

DOI: https://doi.org/10.7554/eLife.43657.011

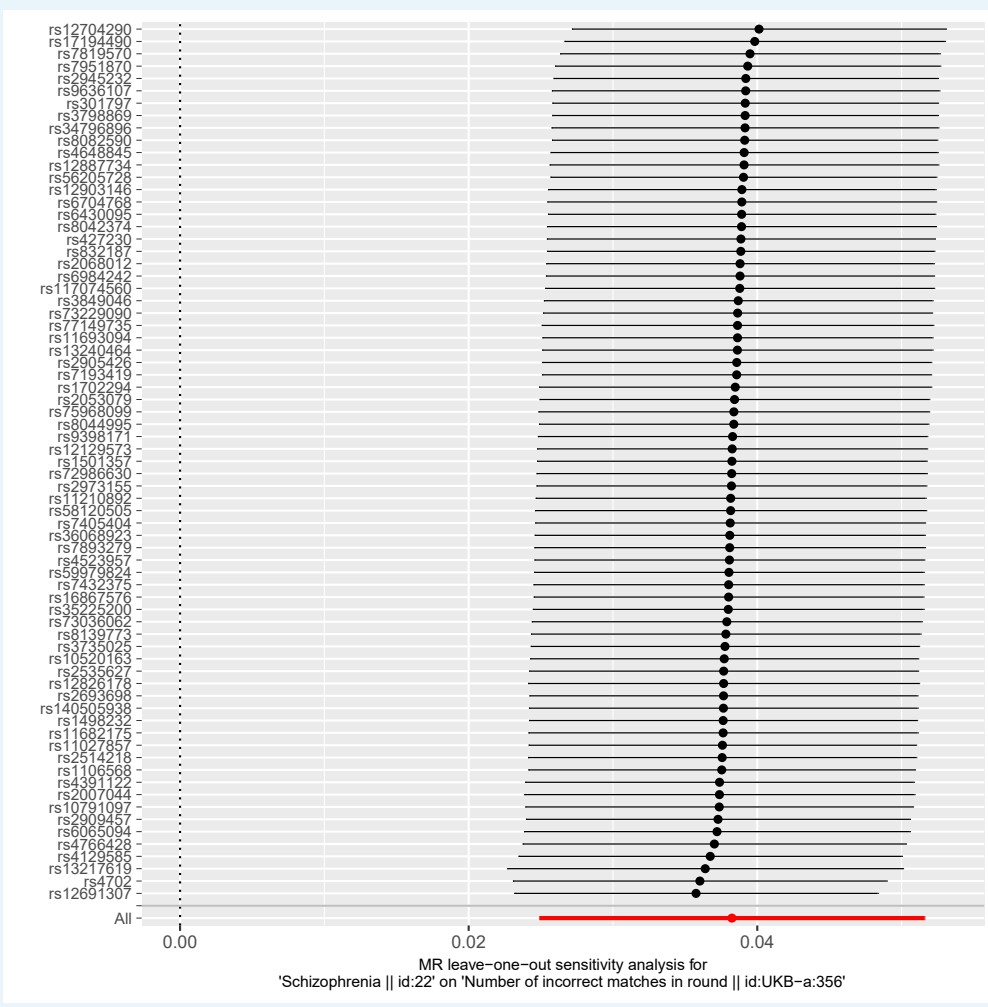

**Appendix 1—figure 2.** A plot illustrating a leave-one out analysis between schizophrenia genetic liability and 'number of incorrect matches in a round'.

DOI: https://doi.org/10.7554/eLife.43657.012

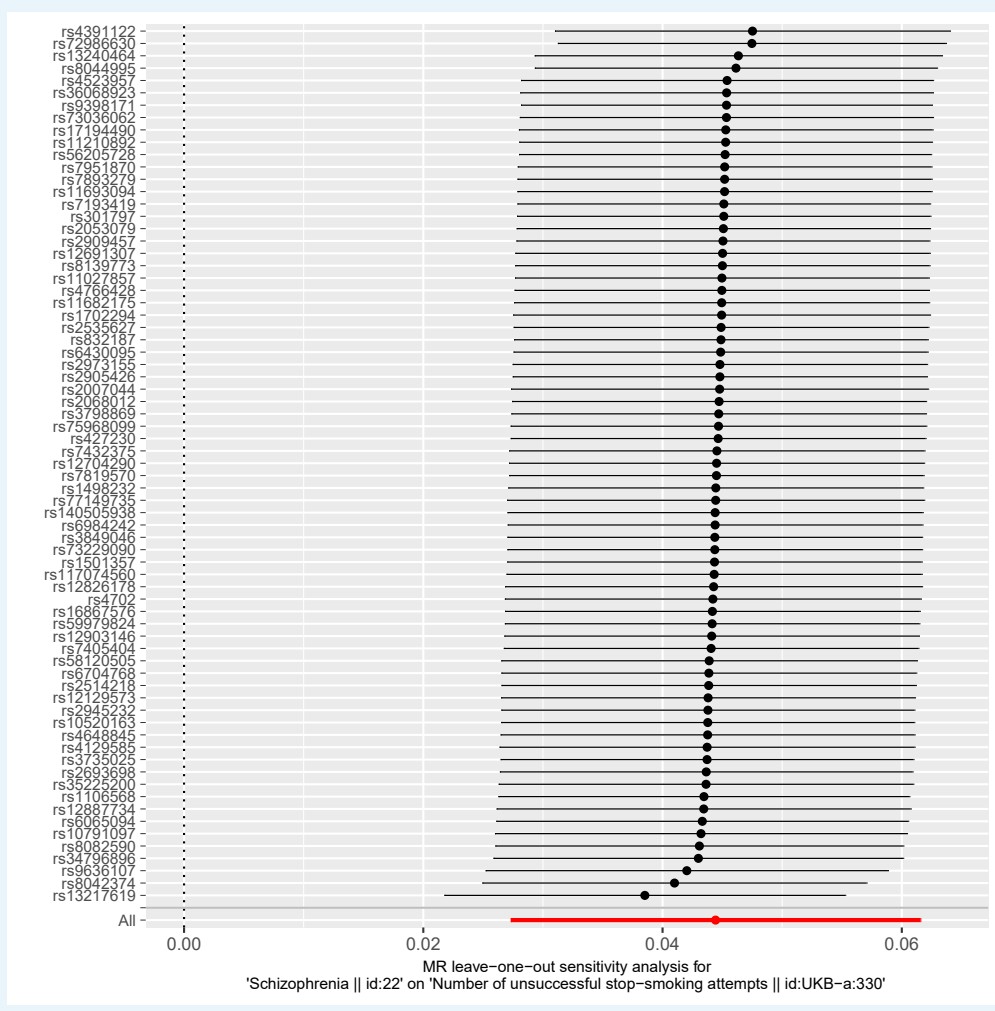

**Appendix 1—figure 3.** A plot illustrating a leave-one out analysis between schizophrenia genetic liability and 'number of unsuccessful smoking attempts'.

DOI: https://doi.org/10.7554/eLife.43657.013

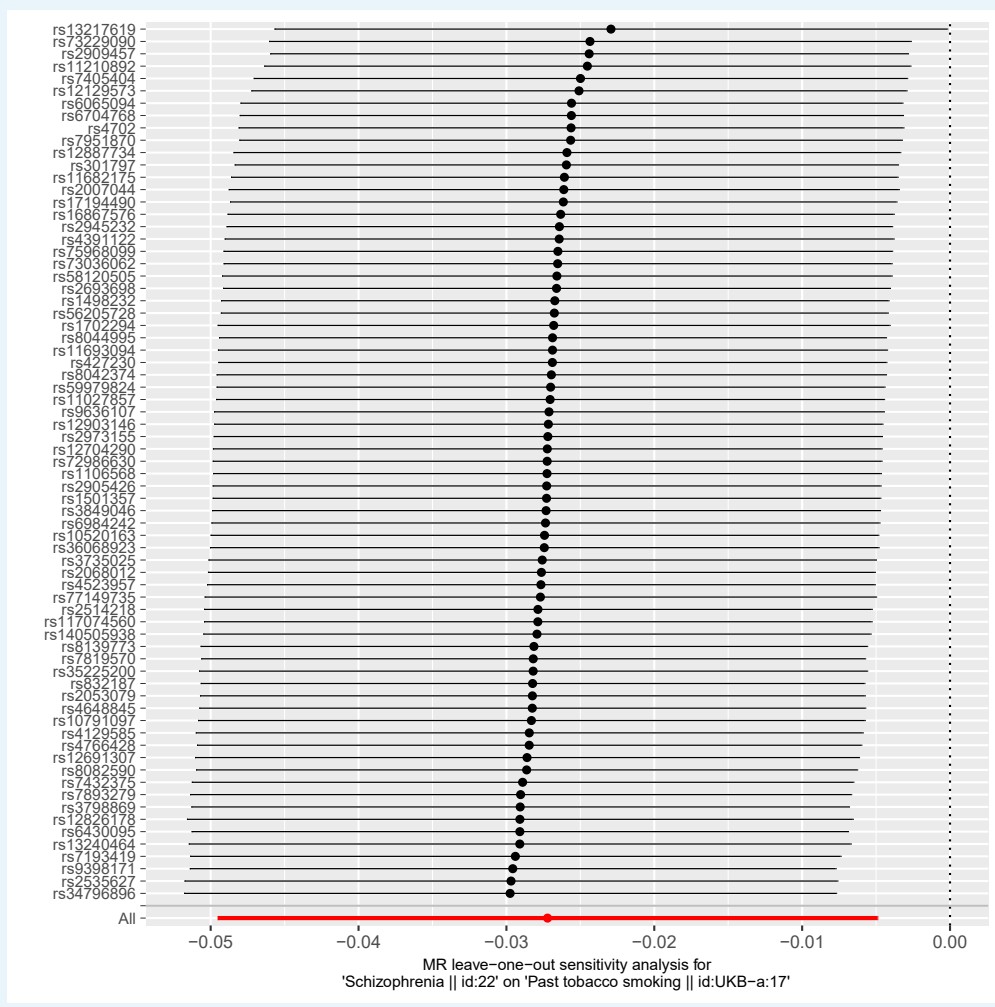

**Appendix 1—figure 4.** A plot illustrating a leave-one out analysis between schizophrenia genetic liability and past tobacco smoking.

DOI: https://doi.org/10.7554/eLife.43657.014

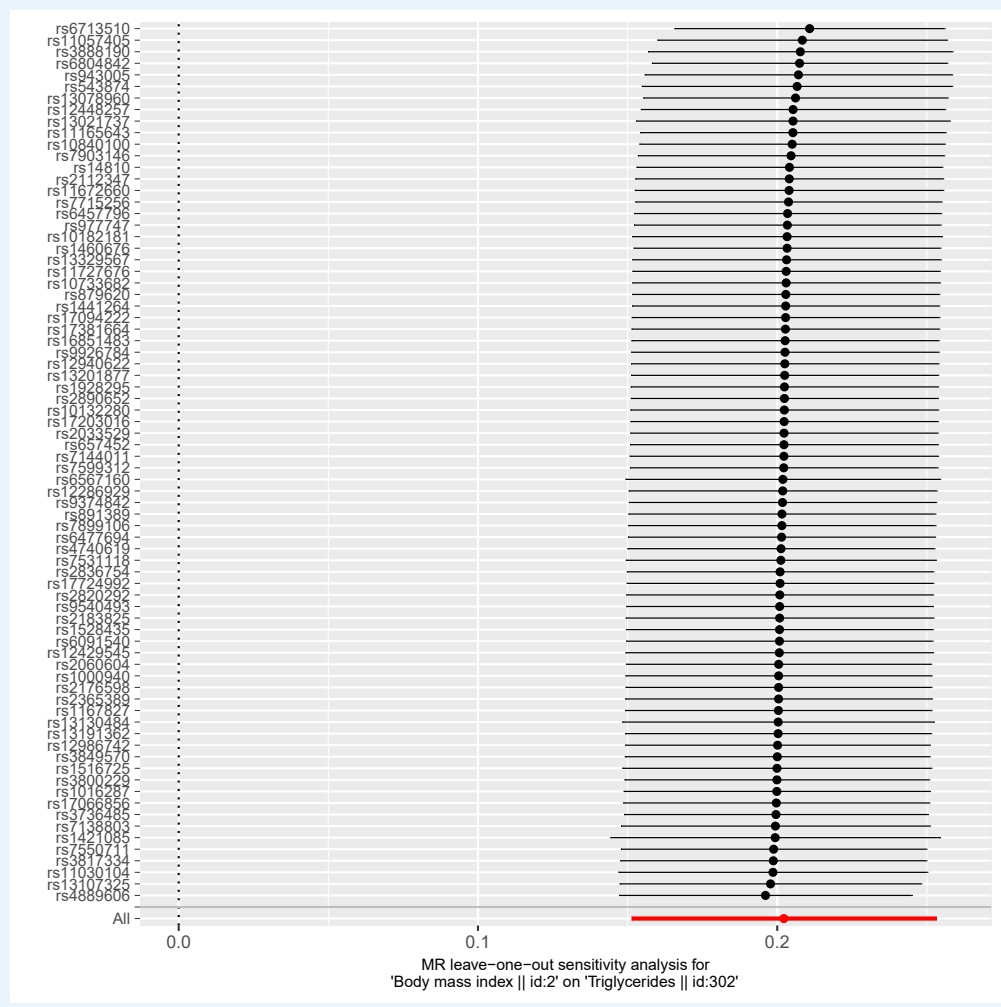

**Appendix 1—figure 5.** A plot illustrating a leave-one out analysis between body mass index and triglycerides.

DOI: https://doi.org/10.7554/eLife.43657.015

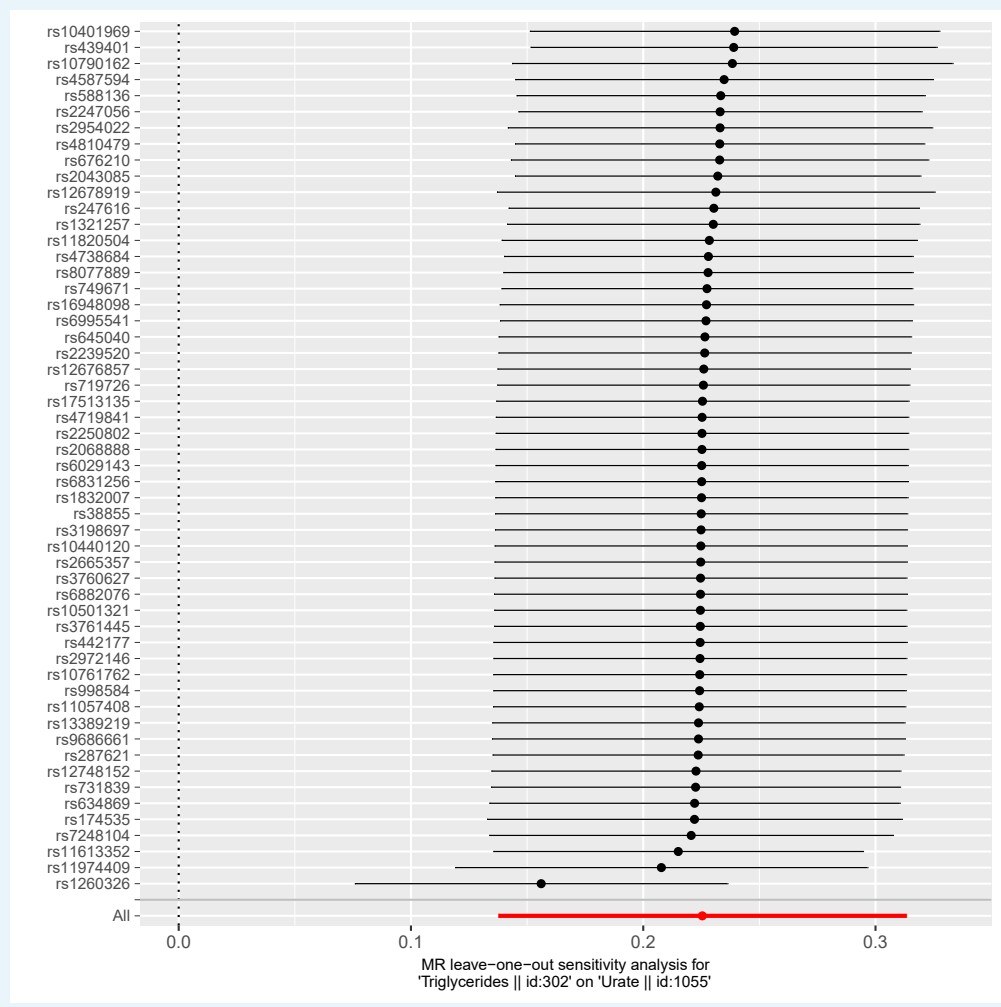

**Appendix 1—figure 6.** A plot illustrating a leave-one out analysis between triglycerides and urate.

DOI: https://doi.org/10.7554/eLife.43657.016

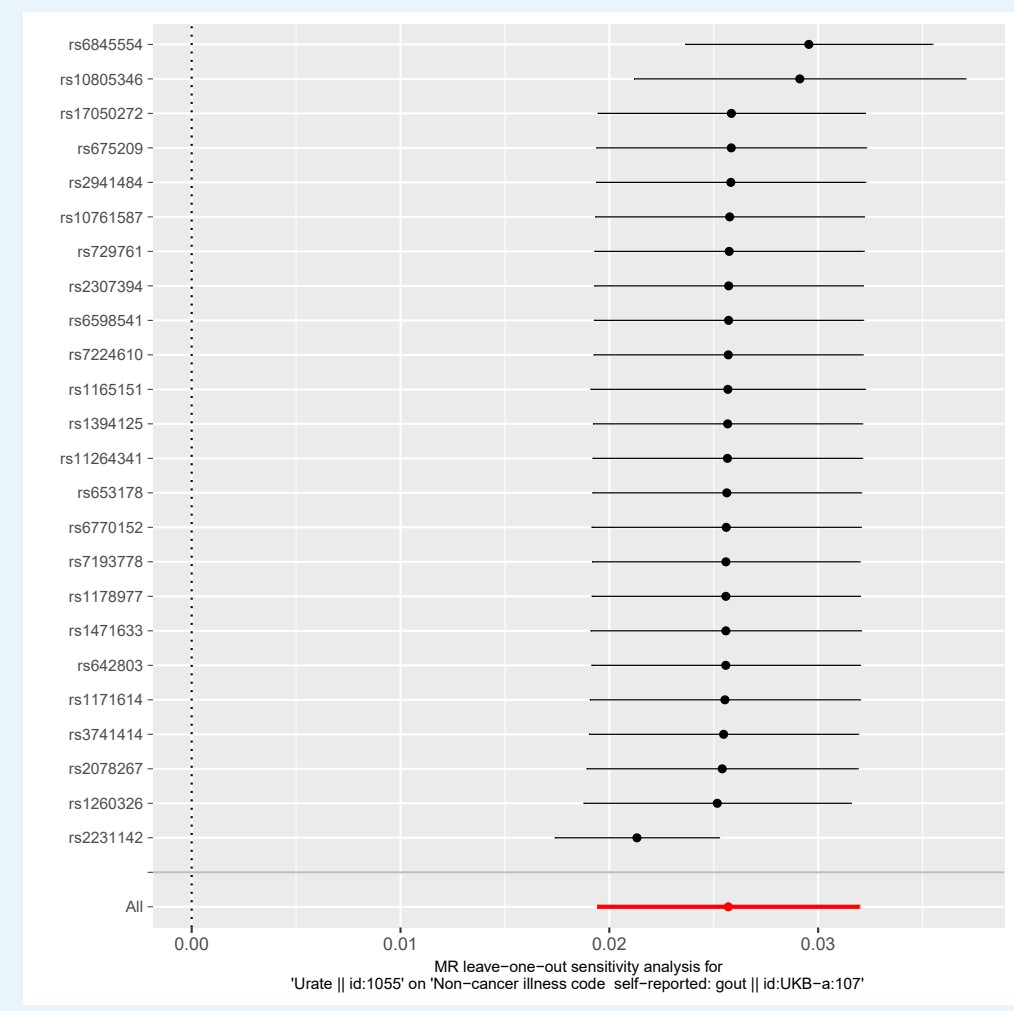

**Appendix 1—figure 7.** A plot illustrating a leave-one out analysis between urate and gout.

DOI: https://doi.org/10.7554/eLife.43657.017

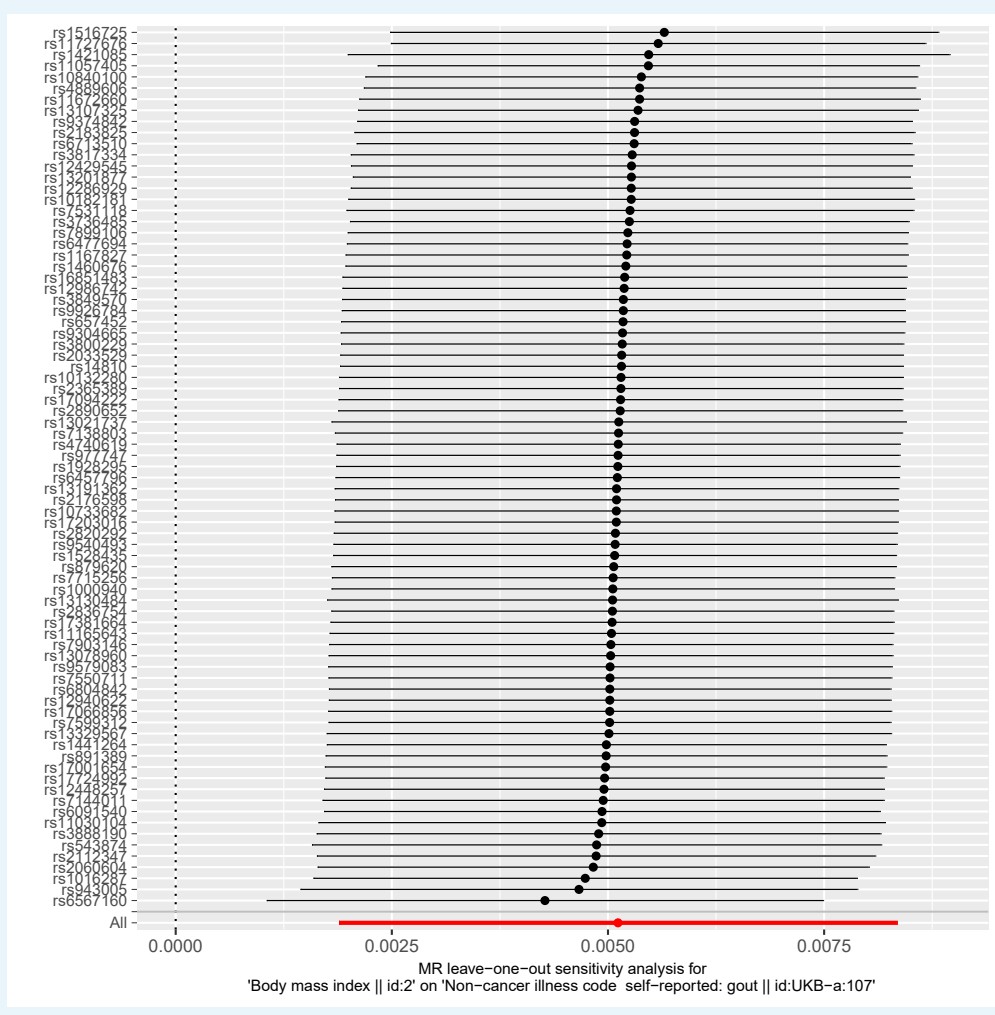

**Appendix 1—figure 8.** A plot illustrating a leave-one out analysis between body mass index and gout.

DOI: https://doi.org/10.7554/eLife.43657.018

