## [Decision Letter]

Thank you for submitting your article "An atlas of polygenic risk score associations to highlight putative causal relationships across the human phenome" for consideration by *eLife*. Your article has been reviewed by three peer reviewers, one of whom is a member of our Board of Reviewing Editors, and the evaluation has been overseen by Mark McCarthy as the Senior Editor. The reviewers have opted to remain anonymous.

The reviewers have discussed the reviews with one another and the Reviewing Editor has drafted this decision to help you prepare a revised submission.

All three reviewers provided a positive review of the paper. However, they have raised several suggestions that should be addressed before the paper can be accepted for publication.

Most of these are pretty minor, and we anticipate should be straightforward for you to address.

Reviewer #2's "major limitation" was discussed among reviewers. The editors clarified that since the paper is being considered as a Tools and Resources article, that this particular concern is not valid and does not need specific attention.

*Reviewer #1:*

This paper describes the atlas of polygenic risk score associations, a very useful resource, made available online through a browser. More than 160 GWAS-derived PRS were created that can be associated with more than 550 traits available in the UK Biobank. The atlas allows uncovering new risk factors of disease and, in combination with the MR-Base browser/software, to test causality.

The paper illustrates how the atlas can be used with a few examples. While generally well written, I do have the following suggestions for the paper:

- Throughout the paper, while the authors do a good job reporting the statistics and various methods, the readers are often left on their own with the final interpretation. E.g. often, the authors will state that there is evidence of a causal relationship, but from the sentence, it is not clear what the exposure and what the outcome is (not everyone will want to check the supplementary tables). It would be good to go through the paper and make sure the causal relationships are described sufficiently.

- Some of the PRSs were not weighed because the GWAS included the "interim release" of UKBiobank. The authors argue that without the weighing, they avoid overfitting of the PRS. It would be good to indeed demonstrate that; i.e. test a weighed PRS on the second part of the UKBiobank that was not part of the GWAS and compare it to the unweighted PRS in the full UKBiobank.

- In the last paragraph of the subsection “An atlas of polygenic risk score associations across the human phenome”, the results for CHD are being discussed. While the ORs are similar as reported before, the AUC-ROC for CAD is much lower than reported in the Khera et al. paper; i.e. 0.60 (here) vs. 0.85 (Khera et al.). Given the high profile of the Khera et al. results it would be worth explaining what this difference might be due to - likely because Khera et al. included age and sex in their model.

- The fact that multiple PRSs reduce the predictive ability is interesting and probably worth a paper on its own. Has this been tested for other outcomes or has this been documented elsewhere? It currently leaves the readers a little on their own without a clear answer to why this might be.

- The authors conclude that schizophrenia may lead to reduced cognitive function. However, in their results they state that there is some evidence for the opposite effect. This may need clarification.

- How did the authors estimate that 77% of the effect of BMI on gout was mediated ?

With reference to the atlas:

- I was surprised to see that a high PRS for adiponectin was associated with high adiposity. I assume that this unexpected positive correlation is because the PRS was calculated based on BMI-adjusted adiponectin. I believe there are other traits that we adjusted (for BMI). It should be made clear in the browser which traits, as it changes the interpretation.

Punctuation throughout the text could be improved.

*Reviewer #2:*

Richardson and colleagues develop an atlas of associations of 162 polygenic scores and 551 traits. They explore this atlas and develop two stories around causal inference for schizophrenia and gout.

Strengths:

1) The atlas may be useful for investigators to explore;

2) There is a web application that will facilitate exploration of the results.

Major limitations:

1) From their exploration of this resource, they highlight two stories. The significance of the biologic insights presented is incremental and limited. As such, should the work be presented as a software or web resource rather than a publication of a scientific observation?

Minor limitations:

1) Discussion can be trimmed and simplified.

*Reviewer #3:*

Richardson et al. present a comprehensive study on an atlas of polygenic risk score associations to highlight putative causal relationships across the human phenome. The authors present a well qualified study pinpointing to the limitations of such an approach for investigating truly causal relationships.

Regarding this manuscript I only have minor comments.

1) Why only 334,398 individuals in the UK Biobank study?

2) Simulation study details can be provided in main text. It was hard to follow what exactly was done in the simulation study.

3) The first paragraph of the subsection “Elucidating risk factors which may play a mediating role along the causal pathway to disease”, contains a speculative statement about how likely it is representative of other findings within the atlas, where the PRS for the disease of interest may not always necessarily be the best polygenic predictor of it. Can you provide another example?

---

## [Author Response]

Reviewer #1:[…] The paper illustrates how the atlas can be used with a few examples. While generally well written, I do have the following suggestions for the paper:- Throughout the paper, while the authors do a good job reporting the statistics and various methods, the readers are often left on their own with the final interpretation. E.g. often, the authors will state that there is evidence of a causal relationship, but from the sentence, it is not clear what the exposure and what the outcome is (not everyone will want to check the supplementary tables). It would be good to go through the paper and make sure the causal relationships are described sufficiently.

Many thanks for this suggestion. We have added detail throughout the paper to clarify which exposures and outcomes are being evaluated when undertaking MR analyses. This includes some of the following additions:

‘Furthermore, in these analyses we model schizophrenia as our exposure within an MR framework with associated complex traits as outcomes (unless stated otherwise).’

‘Along with using MR to investigate the effect of PRS traits on outcomes, we recommend investigating the converse direction of effect where possible (also known as ‘bi-directional’ MR. For example, for the associations detected with the schizophrenia PRS, associated traits in the UK Biobank were modelled as our exposure in an MR setting and schizophrenia was treated as our outcome.’

‘Follow-up MR analyses provided evidence from multiple methods that schizophrenia genetic liability (i.e. our exposure) influences both of these outcomes (Supplementary file 1G).’

‘There was strong evidence that BMI (i.e. our exposure) had a causal effect on each other trait in turn (TG, urate and gout), where effect estimates appeared to be consistent between different MR methods (Supplementary file 1N).’

- Some of the PRSs were not weighed because the GWAS included the "interim release" of UKBiobank. The authors argue that without the weighing, they avoid overfitting of the PRS. It would be good to indeed demonstrate that; i.e. test a weighed PRS on the second part of the UKBiobank that was not part of the GWAS and compare it to the unweighted PRS in the full UKBiobank.

We thank the reviewer for this excellent suggestion. We firstly thought it was important to emphasise in the paper that unweighted PRS cannot completely avoid overfitting, but rather as an approach to reduce this source of bias. As such we have included an additional sentence to clarify this. Furthermore, we have also undertaken the analysis suggested by the reviewer, using the PRS for sleep duration as an example. These results can be found in Supplementary file 1C and demonstrate the problems with overfitting due to overlapping samples, as well as how using an unweighted score in these scenarios can help mitigate this limitation:

‘Of the 162 GWAS we identified, 11 reported that they included UK Biobank participants in their analysis. […] In case they are still useful for follow-up analyses despite overlapping with UK Biobank, these scores have been clearly flagged in Supplementary file 1 by being allocated to the ‘unweighted’ subcategory.’

- In the last paragraph of the subsection “An atlas of polygenic risk score associations across the human phenome”, the results for CHD are being discussed. While the ORs are similar as reported before, the AUC-ROC for CAD is much lower than reported in the Khera et al. paper; i.e. 0.60 (here) vs. 0.85 (Khera et al.). Given the high profile of the Khera et al. results it would be worth explaining what this difference might be due to - likely because Khera et al. included age and sex in their model.

We agree with the reviewer that this would be a valuable addition to the manuscript. We have added the following sentence:

‘A recent study by Khera and colleagues reported a similar odds ratio for CHD in their analysis (OR:>3.0 for the highest 8% of individuals based on their PRS). However, we note that they identified a higher area under curve in their analysis (0.806), which is likely attributed to tuning parameters such as LD clumping, along with covariates adjusted for their analysis.’

- The fact that multiple PRSs reduce the predictive ability is interesting and probably worth a paper on its own. Has this been tested for other outcomes or has this been documented elsewhere? It currently leaves the readers a little on their own without a clear answer to why this might be.

Again we thank the reviewer for this point. We have acknowledged that this work deserves further evaluation by future studies, following on from some possible explanations for the reader to why this result was observed:

‘This could potentially due to the increase in variance incorporated into prediction analyses from scores that do not directly influence CHD, or alternatively may indicate that they are spurious associations. Additional research is required to evaluate the contribution of multiple PRS as predictors of a single outcome. Doing so may help develop a greater understanding into which traits can help predict disease outcomes using PRS.’

- The authors conclude that schizophrenia may lead to reduced cognitive function. However, in their results they state that there is some evidence for the opposite effect. This may need clarification.

Thank you for this suggestion. On the Results section we mentioned that there is ‘weak evidence of a causal effect in the opposite direction’. However, by this we mean that the results suggest that there is unlikely to be an effect in the opposite direction. To add some clarity to this point, we have rephrased this sentence as:

‘In contrast, we did not detect strong evidence of a causal effect in the opposite direction for these associations (i.e. evaluating the effect on measures of cognition and memory on schizophrenia risk), in particular after applying MR directionality filtering and when evaluating results from the weighted median and mode methods (Supplementary file 1H).’

- How did the authors estimate that 77% of the effect of BMI on gout was mediated?

Many thanks for this comment. We have added the following to the Materials and methods section of the paper to help clarify how we calculated the proportion mediated:

*‘*When undertaking our example of mediation MR in this study, we calculated the proportion mediated along the causal pathway from exposure to outcome using effect estimates derived using the IVW method, where:

*Proportion mediated =* directeffect-indirecteffectdirecteffect

The direct effect here is the IVW effect estimate derived for the association between the exposure (i.e. BMI) and our outcome (i.e. gout). The indirect effect was calculated as the product of all IVW effect estimates derived for all relationships along the causal pathway of interest (i.e. the effect of BMI on triglycerides, the effect of triglycerides on urate and the effect of urate on gout).’

With reference to the atlas:- I was surprised to see that a high PRS for adiponectin was associated with high adiposity. I assume that this unexpected positive correlation is because the PRS was calculated based on BMI-adjusted adiponectin. I believe there are other traits that we adjusted (for BMI). It should be made clear in the browser which traits, as it changes the interpretation.

We are thankful to the reviewer for this suggestion. They are correct that the PRS for adiponectin was based on a GWAS which adjusted for BMI. We have therefore added this information to Supplementary file 1 and the web application as recommended.

Punctuation throughout the text could be improved.

Many thanks for this point. One of my co-authors has kindly gone over the manuscript to help improve the punctuation.

Reviewer #2:[…] Major limitations:1) From their exploration of this resource, they highlight two stories. The significance of the biologic insights presented is incremental and limited. As such, should the work be presented as a software or web resource rather than a publication of a scientific observation?

Many thanks for this suggestion. As discussed with the editor, we have now submitted this work as tools and resources paper.

Minor limitations:1) Discussion can be trimmed and simplified.

We thank the reviewer for this point. We have edited sections of the Discussion and subsequently reduced its overall length.

Reviewer #3:[…] Regarding this manuscript I only have minor comments.1) Why only 334,398 individuals in the UK Biobank study?

Thank you for raising this point as we realise it is not clarified in the main text of the paper. As such we have added the following sentence to address this:

‘Our final sample size for analysis consisted of 334,398 individuals. This was determined using a strict exclusion criterion to reduce false positive associations, removing individuals with withdrawn consent, evidence of genetic relatedness or who were not of ‘white European ancestry’ based on a K-means clustering (K=4).’

2) Simulation study details can be provided in main text. It was hard to follow what exactly was done in the simulation study.

Thanks very much for this suggestion. We have added some further details regarding the simulation study to the Materials and methods. Furthermore, we have added a link to the GitHub repository with detailed methods and all the code necessary to run the simulations:

‘Our simulation study concerned two different models; the causal model (simulating a risk factor which has a causal effect on the simulated outcome) and the null model (where there is no causal effect between the simulated exposure and outcome). […] Further details and all the code used to conduct these simulations can be found at https://github.com/explodecomputer/prs-vs-mr.’

3) The first paragraph of the subsection “Elucidating risk factors which may play a mediating role along the causal pathway to disease”, contains a speculative statement about how likely it is representative of other findings within the atlas, where the PRS for the disease of interest may not always necessarily be the best polygenic predictor of it. Can you provide another example?

We thank the reviewer for raising this point, and on reflection feel that this statement may have been a little strong and have therefore toned it down. We have also provided another example where the strongest association for a PRS is a risk factor instead of outcome itself:

“Although many of the PRS in our analysis may be the best polygenic predictors for their target disease/trait, there may be other examples similar to this where the strongest association for an outcome is not the corresponding PRS. For example, the strongest association for birth weight as an outcome in our atlas was with the height PRS (*Β=0.080, SE=0.002, P=1.31x10^-249^).”*